# Assessment of Interference in CIEDs Exposed to Magnetic Fields at Power Frequencies: Induced Voltage Analysis and Measurement

**DOI:** 10.3390/bioengineering12070677

**Published:** 2025-06-20

**Authors:** Mengxi Zhou, Djilali Kourtiche, Julien Claudel, Patrice Roth, Isabelle Magne, François Deschamps, Bruno Salvi

**Affiliations:** 1Institut Jean Lamour, Université de Lorraine-CNRS (UMR 7198), 2 allée André Guinier, Campus Artem, 54000 Nancy, France; djilali.kourtiche@univ-lorraine.fr (D.K.); julien.claudel@univ-lorraine.fr (J.C.); patrice.roth@univ-lorraine.fr (P.R.); 2EDF, Direction Prévention Santé Sécurité Groupe, Pôle Prévention des Risques, 4 rue Floréal, 75017 Paris, France; isabelle.magne@edf.fr; 3RTE, Direction Développement Ingénierie, Département Concertation et Environnement, Place du Dôme, 92073 Paris, La Défense CEDEX, France; francois.deschamps@rte-france.com (F.D.); bruno.salvi@rte-france.com (B.S.)

**Keywords:** magnetic field, power frequency, cardiac implantable electronic device, EMF risk assessment, experimental measurement

## Abstract

Despite ongoing concerns about electromagnetic interference affecting cardiac implantable electronic devices (CIEDs) in the electrical industry workplaces, no study has experimentally assessed induced voltages in CIEDs under exposure to power-frequency magnetic fields. This study addresses this gap by quantifying such interference using a dedicated experimental setup to reproduce high intensity magnetic fields and to measure voltages induced on CIEDs under exposure. A thorough analysis was carried out in comparison with formula-based and simulation approaches applied in previous studies. The induced voltages on CIEDs were measured across varying configurations, including sensing mode, implantation method, exposure frequency, and magnetic field orientation. Our findings reveal the induced voltage levels under exposure from a statistical perspective and highlight correlations between susceptibility and the impact factors, with unipolar configurations and left pectoral implants exhibiting the highest susceptibility. This work provides insights into electromagnetic interference risks for CIED carriers and supports the development of individual protection strategies to enhance occupational safety.

## 1. Introduction

One of the most significant contemporary treatments for cardiovascular diseases is the use of cardiac implantable electronic devices (CIEDs). Approximately 600 pacemakers (PMs) per million people are implanted in European countries, with rates exceeding 1000 per million in countries such as Italy, Sweden, France, Portugal, and Lithuania. Meanwhile, the number of implantations for implantable cardioverter defibrillators (ICDs) and cardiac resynchronization therapy (CRT) devices have been increasing in recent years [1]. The risk factors for cardiovascular diseases have increasingly affected younger populations due to modern lifestyle choices, including diet, alcohol consumption, tobacco use, and physical activity. Accordingly, evaluating the potential risks of electromagnetic interferences (EMIs) to CIED carriers is essential not only in everyday scenarios but also in occupational environments for those who are professionally active [2,3,4].

A typical context is the workplace near electrical networks and appliances such as power lines, substations, and transformers in the electricity industry, where extremely low frequency (ELF) electric and magnetic fields may be significantly higher than the public exposure levels defined by the international standards and recommendation [5]. CIEDs are typically equipped with narrow bandpass filters to exclude non-cardiac signals. However, the frequency range of human cardiac signals typically falls within 0.05 and 150 Hz, encompassing the 50 and 60 Hz power frequencies. Consequently, workers in the electrical industry who have CIED implants may encounter potential risks from EMI.

Directive 2013/35/EU established limits to protect individuals from adverse health effects caused by overexposure [6]. According to ICNIRP Guidelines [7], the reference exposure levels are 0.2 mT for public exposure and 1 mT for occupational exposure. Based on the compliance documents, investigations were conducted across various contexts. An in vivo provocation study of 50 Hz electric and magnetic fields (EMFs) was conducted among fifteen patients with CIEDs. Some interferences were detected when patients were exposed to fields below the reference exposure levels, but only under worst-case conditions [8]. In the in vivo study involving twenty-four volunteers with CIEDs, participants with unipolar programmed pacemakers experienced notable interference when exposed to the ICNIRP reference level for occupational exposure, while minor abnormalities were observed in electrocardiography (ECG) readings for those with bipolar settings [9]. Additionally, in vivo studies on individuals may reflect specific cases and conditions. The use of bipolar leads, Tip–Ring spacing and location, sensitivity settings, exposure orientation, and even biological characteristics of the implant recipient affect the susceptibility of CIEDs to exogenous EMI [10,11,12]. Although preliminary research has indicated potential interactions between power–frequency electromagnetic fields (EMFs) and CIEDs, few experimental investigations have been conducted to establish a comprehensive analysis. The physical quantities underlying the interference in CIEDs remain insufficiently documented.

A thorough investigation has been conducted to assess the effect of PMs and ICDs exposed to high-intensity electric fields (EFs) at 50 Hz [13]. Based on the findings, a risk assessment method for workers with CIEDs under occupational EF exposures was proposed accordingly [14]. In this study, we focus on magnetic field (MF) exposures with two primary objectives. First, we aim to visualize and quantify the interference of power-frequency MFs on CIEDs through experimental approaches. Second, we investigate the impact of exposure variables, including implantation method, orientation of polarization, and frequency, and quantify these factors. By enhancing our understanding of EMI on CIEDs under MF exposure, this paper intends to conduct a thorough analysis to ensure patient safety and device reliability. In addition, the findings may contribute to establishing effective EMF protection and preventive measures for workplace safety.

## 2. Materials and Methods

Typically, a CIED is composed of an impulse generator (IPG), containing the battery and electronic circuit, and one or more leads equipped with electrodes that detect the heart rate and deliver electrical impulses or shocks as therapy when necessary. Depending on patient-specific conditions and physician’s preference, CIED implantation is commonly performed in the right pectoral or left pectoral region. IPG is placed under the skin near the collarbone, and the lead is inserted into a vein and guided to the appropriate heart chamber. The implantation of the device forms an open circuit loop within the human body, while the surrounding conductive biological tissues close this loop. When exposed to MF, a current is induced in this loop, flowing from the electrode(s) back to the IPG through the conductive body tissue (Figure 1). This current is considered a potential hazard to the device, as it generates a voltage at the IPG input, which may cause interference with the electronic system and may result in malfunctions.

In a conservative approach, for a loop exposed to a time-varying homogeneous magnetic field orthogonal to the loop plane, the induced voltage V can by calculated by applying the induction law:V = B × A × 2 × π × *f*(1)
where B is the magnetic flux density (in T), A is the area of the induction loop (in m^2^), and *f* is the frequency. According to this equation, determining the induced voltage depends on both the exposure characteristics and the surface of the induction loop. The calculation of induced voltage can be directly applied to a homogeneous MF oriented perpendicular to the cardiac implant’s induction surface at power frequency (50/60 Hz), which we defined as the default exposure. In this work, to obtain consistent results and to compare the experimental measures with the theoretical calculation and numerical simulations, we performed research exclusively on homogeneous MFs. The employment of homogeneous fields with sinusoidal waveforms enables the establishment of various exposure configurations for subsequent applications. According to the European standard EN 50527-2-1:2017 [15], the maximum realistically achievable effective surface enclosed by a sensing lead and the current path in body tissue is 225 cm^2^ in left pectoral implantation and 135 cm^2^ for right pectoral implantation (deduced by a factor of 0.6). In this work, we studied the configuration with a surface of 226 cm^2^ formed by a semicircle with a diameter of 24 cm to represent the left pectoral implantation, and a configuration with a surface of 157 cm^2^ formed by a semicircle with a diameter of 20 cm to represent the right pectoral implantation.

In the unipolar configuration, the surface enclosed by the current path from the lead electrode Tip to the IPG and the sensing lead represents the unipolar effective induction area *A_Tip_* (Figure 2A). In the bipolar configuration, the cardiac implant detects the heart signals using two electrodes at the end of the sensing lead, namely Tip and Ring. Each electrode establishes an induction area, *A_Tip_* and *A_Ring_*, with the IPG serving as a common reference. The induction surface is defined by the difference between these two areas (*A_Tip_* − *A_Ring_*). Consequently, the effective induction area in bipolar configuration is enclosed by the Tip–Ring pair and the respective current paths from each electrode to the IPG (Figure 2B). A plastic lead fixing plate was fabricated to guide the sensing leads to form proper induction surfaces (Figure 2C).

Under a given MF exposure and with knowledge of the induction surface, the voltage induced on the cardiac implant can be calculated using Equation (1). However, this solution remains theoretical. To further investigate the effects of MF exposure on cardiac implants, we conducted an experimental evaluation to quantify the MF-induced interference by measuring induced voltage on the cardiac implant during exposure. Helmholtz coils with a diameter of 520 mm were used as the exposure source to generate a homogeneous MF oriented horizontally. High currents were supplied to the coils to achieve MF levels of up to 2 mT. A PVC container (L × W × H: 310 mm × 170 mm × 220 mm) filled with saline solution, with an electrical conductivity of 0.18–0.22 S/m, was used as a phantom to simulate human body tissue [16]. In our previous study, a dedicated system was designed and implemented to measure the induced voltage on a cardiac implant during exposure [13]. While maintaining its essential functions, the circuit size was minimized to fit within a shielding housing similar in size to ICDs (65 mm × 50 mm). It serves as a substitute for the cardiac implant, sensing signals in the manner of actual cardiac implants, while transmitting the measured data in real time via optical fiber. A standard IS-1 sensing lead with a Tip–Ring spacing of 10 mm was employed. To ensure the compatibility between the coil dimensions and the phantom size and to confirm that the measurement device is properly exposed within a homogeneous field, the MF distribution was verified through simulation using CST Studio Suite^®^ (Dassault Systèmes, Vélizy-Villacoublay, France). The phantom is located in a region where the MF remains uniform, with a relative deviation of 3% (Figure 3).

To conduct a thorough investigation among individual physiological structures and parameters of devices and their implantations, we performed measurements across various factors, including sensing mode, implantation method, frequency, and exposure orientation. Here, *θ* represents the angle between the field direction and the normal line to the induction surface, while frontal exposure (*θ* = 0°) was defined as the default MF orientation (Figure 3). During the measurements, exposure levels were adjusted for different configurations, and corresponding induced voltages were recorded. Although linearity can be theoretically predicted, we collected 12 data points for each configuration, with each point comprising 32 signal samples. Due to the high sensitivity of the measuring system, these results were post-processed to eliminate interference from unwanted sources at other frequencies. Simulation using CST Studio^®^ for the overall experimental setup was conducted for a typical case to compare the experimental measurements to numerical simulations left pectoral implantation, frontal exposure, and 50 Hz exposure. The modeling of the device under test, the sensing lead, and the exposure system in the simulation respect the actual dimensions and materials of the experimental setup. An electrical conductivity of 0.2 S/m was used for phantom solution in the simulation in accordance with the average conductivity of the body in ICNIRP Guidelines.

## 3. Results

### 3.1. Effects of Factors

Figure 4 presents the induced voltages measured in the experimental setup under difference impact factors, including the CIED implantation method, the CIED sensing configuration, the exposure frequency, and the exposure orientation. Each color represents a specific combination of impact factors. The results exhibit linearity with the exposure level, for both unipolar and bipolar configuration while those in bipolar sensing configuration show a greater dispersion, also due to smaller values. Normalized induced voltage was defined to represent the voltage induced on the CIED when exposed to 1 μT MF. In unipolar sensing, a left-implanted CIED exposed to a frontal MF at 60 Hz is subjected to the highest interference among the configurations, while in bipolar sensing, the same configuration exhibits maximum interference as well. Right pectoral implantation results in the lowest induced voltage levels in the unipolar sensing configuration, whereas in bipolar sensing configuration, those with an angle to the MF orientation exhibit the lowest levels.

### 3.2. Validation of Formula-Based Approach

We conducted further analysis on the left pectoral implantation with maximum achievable effective induction surface, as a typical case. In the bipolar sensing configuration, the Tip–Ring pair was consistently positioned perpendicular to the path from the pair’s center to the IPG to achieve the maximum induction area. In our case, this path measured 16 cm, so the bipolar induction area could be approximately calculated as 8 cm^2^. Table 1 presents a comparison among the normalized induced voltage obtained by calculation via Equation (1), simulation using CST Studio^®^, and experimental measurements for a CIED in the left implantation configuration when exposed to frontal MF at 50 Hz. The results show good overall agreement in scale, while those obtained from the formula-based approach demonstrate a higher level compared to the other two methods.

## 4. Discussion

The findings provide analytical assessment of EMI for CIEDs exposed to MFs within the context of an occupational exposure at power frequencies. It should be emphasized that variations in exposure frequency, implantation configuration, phantom and sensing lead selection may significantly affect the findings from a statistical perspective. In Figure 4, a noticeable level of dispersion may be observed in the results, particularly in bipolar sensing mode. Figure 5 illustrates the real-time observation of measurements for a MF of 1290 μT at 50 Hz. Given the weak level of induced voltages in bipolar sensing mode, the measuring device was configured with an amplification gain of 25; however, the induced voltage still remained at a very low level (Figure 5). This process may also amplify noise, which can significantly affect the measurements in bipolar mode. Simultaneously, the measurement device offers high sensitivity, ensuring that the collection of voltages induced by the exposure may also capture undesired components. Nevertheless, the results provided us with the reliable verification of the impact of the various factors and insight into the levels of induced voltage.

As indicated in the European standard EN 50527, the application of bipolar sensing configuration may reduce EMI by approximately a factor of 10 compared to unipolar sensing. The impact of the IPG could not be assessed on the determination of the induction area. Our measurements revealed a significantly higher ratio for all configurations. For a left pectoral implanted device exposed to a frontal MF at 60 Hz, a bipolar/unipolar ratio of 47 was found to be the lowest of all configurations. While the standard provides a robust level of protection for overall conditions, the margin in bipolar sensing may warrant further discussion for specific cases.

In the investigation of the typical case, the induced voltage calculated using Equation (1) is observed to be slightly higher than those obtained from the two other methods in both sensing configurations. We assume that, as in most studies, the influence of the IPG is typically neglected in the formula-based approach, especially for unipolar sensing mode. In this work, the actual unipolar induction area is smaller than that of the semicircle. In addition, the current path in the conductive medium is distributed over a volume rather than being confined to a one-dimensional line. The spread of the current return path within the volume can be easily deformed or displaced by external influences, potentially reducing the effective induction area. In the bipolar sensing mode, where the two current paths are located in close proximity, this effect may be more pronounced. The impact of exposure orientation is more substantial in bipolar sensing mode. This likely occurs because the projected area of the angled configuration becomes too small to prevent significantly influence from this effect in the conductive medium. In addition, the phantom solution’s induction manifested as a capacitive effect during exposure and it is not considered in the conservative approach [17]. To mitigate its influence on the results, we implemented a filtering module in the measurement circuit. Nevertheless, residual interference remained challenging to quantify precisely. This important observation warrants further investigation.

Our measurement in bipolar sensing mode (0.10 μV per 1 μT exposure) is in good agreement with findings for the anatomical model implanted with a ICD (0.099 μV per 1 μT exposure), where only bipolar mode applies [18]. Based on our experimental findings, it can be deduced that, in the typical case, 20 μV may be induced on the CIEDs at the reference level for public exposure specified in the ICNIRP Guidelines (0.2 mT) and 100 μV at the reference level for occupational exposure (1 mT). In the previous assessment of induced voltages in CIEDs exposed to electric fields, no dysfunction was observed below a threshold of 130 μV [14]. Therefore, we may infer that there is no substantial hazard within the reference levels defined in this guideline. Further investigation involving a broader range of devices exposed to MFs would be necessary to confirm this conclusion.

## 5. Conclusions

This study carried out experimental measurements for MF-exposed CIEDs at power frequencies and validated previous findings obtained through computational and numerical approaches. The results provide a distinct analysis on various impact factors related to exposure and CIED itself. Using the conservative formula-based approach, general protection can be ensured. In addition, with appropriate selection of device technology and medical techniques, a safety margin specific to certain configurations may be proposed and further investigated.

## Figures and Tables

**Figure 1 bioengineering-12-00677-f001:**
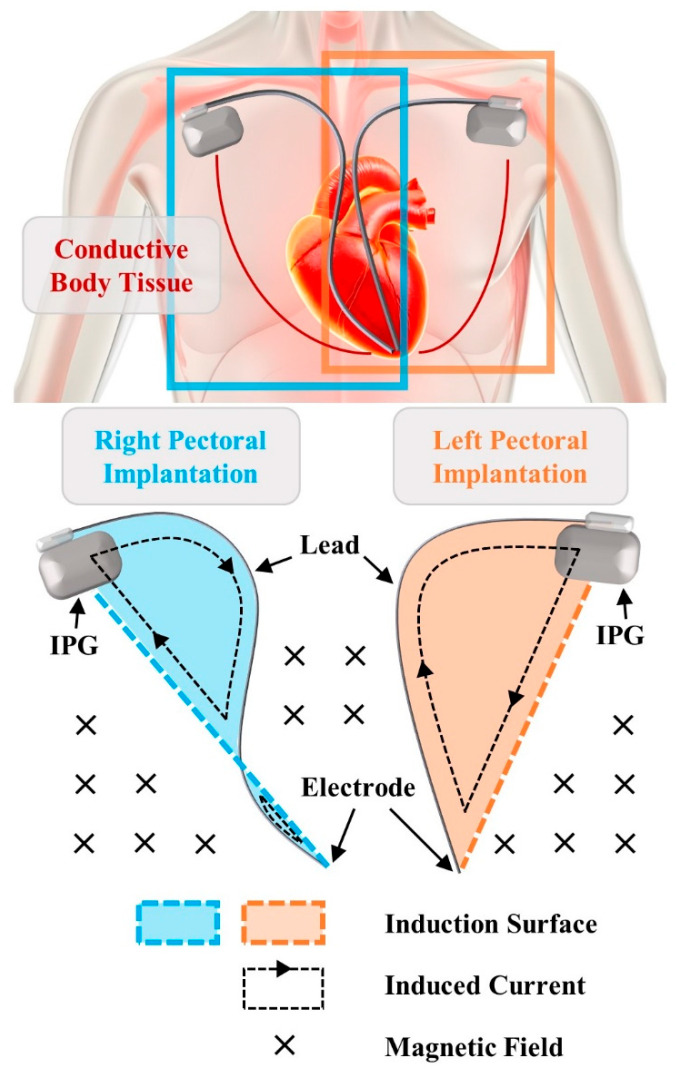
Induced currents in right and left pectoral implantations under MF exposure.

**Figure 2 bioengineering-12-00677-f002:**
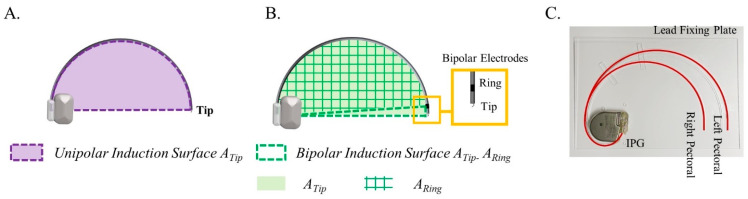
(**A**) Induction surface for unipolar (**B**) Induction surface for bipolar mode (**C**) Implantation interpretations.

**Figure 3 bioengineering-12-00677-f003:**
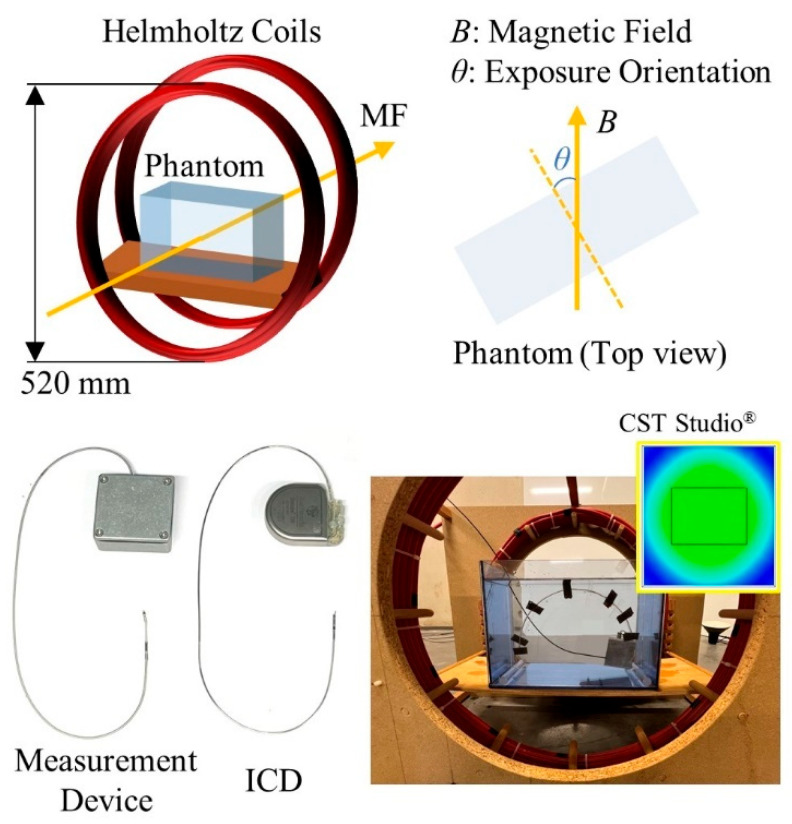
Experimental configuration and measurement visualization.

**Figure 4 bioengineering-12-00677-f004:**
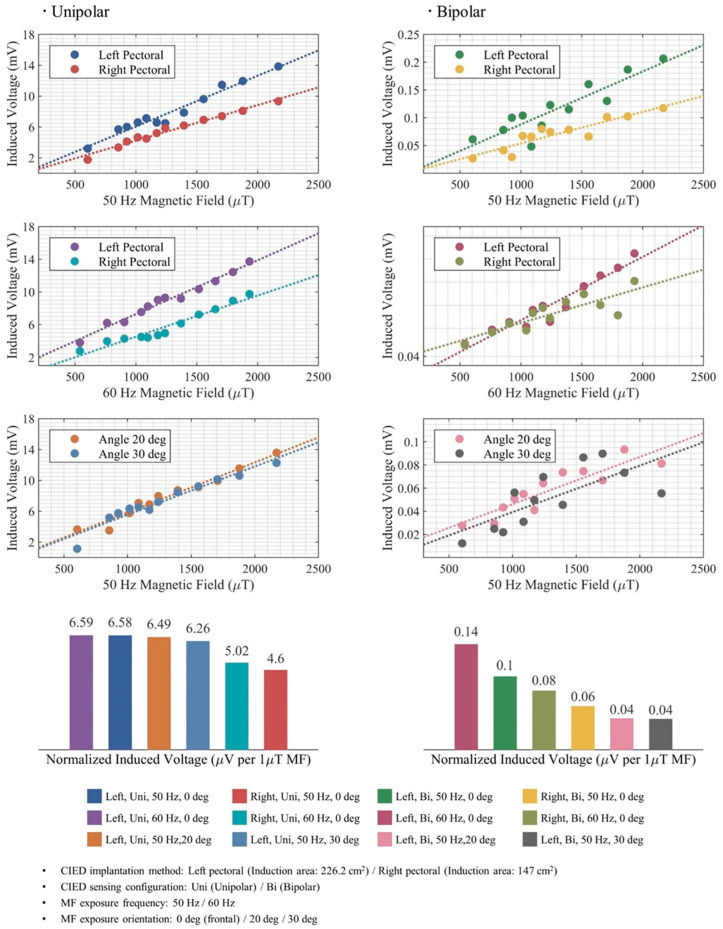
Induced voltages in CIEDs due to MF exposure in different configurations.

**Figure 5 bioengineering-12-00677-f005:**
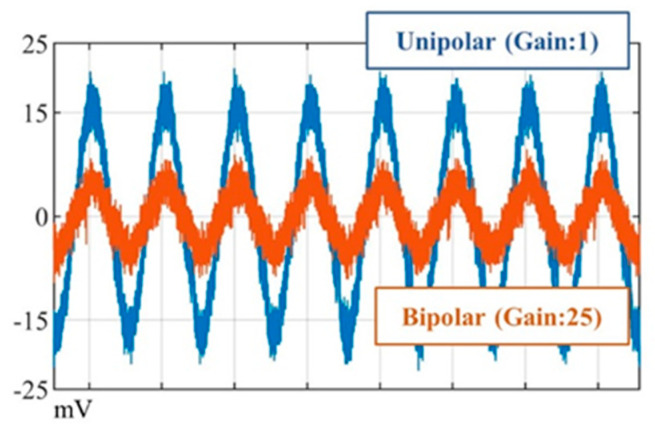
Induced voltages measured for a MF of 1290 μT at 50 Hz.

**Table 1 bioengineering-12-00677-t001:** Result summary for a CIED in left implantation configuration exposed to 1 μT MF at 50 Hz.

Induction Area		Induced Voltage					
		Calculation		Simulation		Measurement	
Unipolar	Bipolar	Unipolar	Bipolar	Unipolar	Bipolar	Unipolar	Bipolar
cm^2^	cm^2^	μV	μV	μV	μV	μV	μV
226.20	8.00	7.10	0.25	6.48	0.16	6.58	0.10

## Data Availability

The research data associated with this article are included within the article.

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
