# Peer review of "Assessment of Interference in CIEDs Exposed to Magnetic Fields at Power Frequencies: Induced Voltage Analysis and Measurement"

_bioengineering, 2025, doi:10.3390/bioengineering12070677_

Round 1
Reviewer 1 Report
Comments and Suggestions for Authors
The paper provides original experimental results, along with theoretical results and numerical simulation results., unlike most of the published papers in this field. The results reported will be of help in this field. In the opinion of this reviewer the paper lack of details in the description of the methods. Authors are invited to improve the description of the methods, as suggested below:
Major issues:
Please provide a more accurate description of the experimental set-up and simulations, and in particular:
- In the experimental set-up, was a commercial lead used? If so please report manufacturer and model, otherwise please provide details on the lead/wire main construction parameters. What was the interbipole distance?
- CST simulation were implemented to assess the induced voltages and to compare with experimental measurement and theoretical calculation. Please provide details on the simulation parameters (e.g. mesh resolution, lead modelling, coil modelling)
These details are relevant to allow other researcher to replicate the results or to compare the results with previous or future works.
Minor issues:
Introduction – row 45 : Avoid the use of the general term “high”. Consider to use a more specific term (i.e. significantly higher than the general population exposure levels defined by international standards and recommendation)). Please check for similar issues in other sections of the manuscript.
Materials and Methods rows 95-98: Please specify that the equation is valid only under the assumption of Homogeneous Magnetic field, orthogonal to the plane of the lead….
Materials and Methods row 101. “reproductivity” please rephrase: eg: “To obtain consistent results and to compare the experimental measures with the theoretical calculation and numerical simulations”
Materials and Methods row 104-110. Authors refer to 225 cm as a “maximum realistically achievable surface enclosed by a sensing lead and the current path”. Please note that according to the 50527-1 this is an effective induction area and not a geometrical area. Authors used a geometrical area of 226 cm for left implantation, please clarify/discuss (also because the dimension of coils and trunk simulator used may affect the effective induction area).
Materials and Methods – fig. 3: Simulation were performed to evaluate the magnetic field homogeneity, results are reported as a color chart, without a scale/range, please report the obtained results in terms of maximum inhomogeneity within the induction area of interest.
Discussion: Please acknowledge/discuss that the impact (if any) of current induced in the saline were not considered in the theoretical calculation (see Scholten A, Silny J. The interference threshold of unipolar cardiac pacemakers in extremely low frequency magnetic fields. J Med Eng Technol. 2001 Sep-Oct;25(5):185-94. doi: 10.1080/03091900110066419. PMID: 11695658.) Was the contribution of these currents to the induced voltage (if any) considered in the simulations?
Comments on the Quality of English LanguageMostly adeguate. Minor changes may improve the manuscript.
Author Response
Dear Reviewer,
Thank you for the comments on this article! We are grateful for your valuable and inspiring comments.
We have revised the manuscript based on your feedback. The original questions and our responses are listed in below. Please feel free to contact us if you need any further information.
Best regards
The paper provides original experimental results, along with theoretical results and numerical simulation results., unlike most of the published papers in this field. The results reported will be of help in this field. In the opinion of this reviewer the paper lack of details in the description of the methods. Authors are invited to improve the description of the methods, as suggested below:
Major issues:
Please provide a more accurate description of the experimental set-up and simulations, and in particular:
- In the experimental set-up, was a commercial lead used? If so please report manufacturer and model, otherwise please provide details on the lead/wire main construction parameters. What was the interbipole distance?
- CST simulations were implemented to assess the induced voltages and to compare with experimental measurement and theoretical calculation. Please provide details on the simulation parameters (e.g. mesh resolution, lead modelling, coil modelling)
These details are relevant to allow other researcher to replicate the results or to compare the results with previous or future works.
In the experimental set-up, it was an IS-1 sensing lead with a Tip-Ring spacing of 10 mm. Modeling of the sensing lead, the device under test, and the exposure system in the simulation respect the actual dimensions and materials of the experimental set-up. These details have been added and highlighted in the revised manuscript.
Minor issues:
Introduction – row 45 : Avoid the use of the general term “high”. Consider to use a more specific term (i.e. significantly higher than the general population exposure levels defined by international standards and recommendation)). Please check for similar issues in other sections of the manuscript.
Revised in the manuscript.
Materials and Methods rows 95-98: Please specify that the equation is valid only under the assumption of Homogeneous Magnetic field, orthogonal to the plane of the lead….
Revised in the manuscript.
Materials and Methods row 101. “reproductivity” please rephrase: eg: “To obtain consistent results and to compare the experimental measures with the theoretical calculation and numerical simulations”
Revised in the manuscript.
Materials and Methods row 104-110. Authors refer to 225 cm as a “maximum realistically achievable surface enclosed by a sensing lead and the current path”. Please note that according to the 50527-1 this is an effective induction area and not a geometrical area. Authors used a geometrical area of 226 cm for left implantation, please clarify/discuss (also because the dimension of coils and trunk simulator used may affect the effective induction area).
Accroding to the 50527, the maximum achievable geometrical area would be 398 cm2 and the maximum realistically achievable effective induction area is 225 cm2. The induction area for left implantation in the standard took 225 cm2 for unipolar leads, corresponding to our typical case with an effective induction area of 226 cm2. The inaccurate term has been revised to “maximum realistically achievable effective induction area”.
Materials and Methods – fig. 3: Simulation were performed to evaluate the magnetic field homogeneity, results are reported as a color chart, without a scale/range, please report the obtained results in terms of maximum inhomogeneity within the induction area of interest.
The color chart illustrates the magnetic field exposure distribution, while the rectangle represents the phantom boundary. We analyzed field homogeneity within the phantom region where contains the measurement device and quantified its relative deviation (highlighted in the manuscript).
“The phantom is located in a region where the MF remains uniform, with a relative deviation of 3% (Fig.3).”
Discussion: Please acknowledge/discuss that the impact (if any) of current induced in the saline were not considered in the theoretical calculation (see Scholten A, Silny J. The interference threshold of unipolar cardiac pacemakers in extremely low frequency magnetic fields. J Med Eng Technol. 2001 Sep-Oct;25(5):185-94. doi: 10.1080/03091900110066419. PMID: 11695658.) Was the contribution of these currents to the induced voltage (if any) considered in the simulations?
Thank you for this insightful question and reference. Indeed, we encountered this phenomenon in our measurements, where the phantom solution's induction manifested as a capacitive effect. To reduce its influence on our results, we implemented a filtering module in the measurement circuit. Nevertheless, its impact remained challenging to quantify precisely. This important observation warrants further investigation, and we have addressed it in the Discussion section in the revised manuscript. (highlighted)

Reviewer 2 Report
Comments and Suggestions for Authors
1. In evaluating the potential risk of CIEDs (Cardiac Implantable Electronic Devices) due to power-frequency (50/60 Hz) electromagnetic fields, which is the primary focus of this study, it would be helpful to provide practical examples of environments where such hazardous conditions may realistically occur. Although this work analyzes the effects of a uniform EM field at power frequency, we believe it is more appropriate to identify and model high-intensity EM environments at these frequencies that pose an actual risk in practice.
2. In real-world conditions, CIEDs are in direct contact with conductive tissue within the body, forming a loop as described by the author, which allows voltage to be induced according to the theory presented in Section II. However, the measurement setup shown in Fig. 3 does not appear to replicate this environment adequately. The authors should provide detailed information on the setup, including the material of the phantom case, the type of fluid used inside, and the positioning and orientation of the device. Furthermore, the authors should discuss how these setup parameters might introduce discrepancies or errors compared to actual physiological conditions. The reviewer questions whether this setup can accurately measure the induced voltage as predicted by Equation (1).
3. As shown in Fig. 4, there is almost no noticeable difference between the results at angles of 20° and 30°, and no clear trend is observed. This lack of variation warrants further discussion. Additionally, the authors should clarify the rationale behind choosing only 20° and 30° as test angles. If the goal was to comprehensively assess angular dependence, a wider range of angles (e.g., 20°, 30°, 60°, 90°) should have been included.
4. As noted in question (2), it remains questionable whether the measured and simulated values in this study follow the same trend as the calculated values. According to Table 1, there is a substantial discrepancy between the calculated results and the simulated/measured data. Given the simplicity of the tested pathway, such significant deviations further reinforce the reviewer’s concerns regarding the validity of the theoretical assumptions and the accuracy of the measurement setup.
Author Response
Dear Reviewer,
Thank you for the comments on this article! We are grateful for your valuable and inspiring comments.
We have revised the manuscript based on your feedback. The original questions and our responses are listed in below. Please feel free to contact us if you need any further information.
Best regards!
- In evaluating the potential risk of CIEDs (Cardiac Implantable Electronic Devices) due to power-frequency (50/60 Hz) electromagnetic fields, which is the primary focus of this study, it would be helpful to provide practical examples of environments where such hazardous conditions may realistically occur. Although this work analyzes the effects of a uniform EM field at power frequency, we believe it is more appropriate to identify and model high-intensity EM environments at these frequencies that pose an actual risk in practice.
As this study aims to contribute to the development of the European Standard for occupational safety of CIED carriers, particularly in workplaces within the electric industry, the selection of the exposure characteristics is critical to ensuring the study’s reproducibility. Therefore, we employed homogeneous fields with sinusoidal waveforms, enabling the establishment of various exposure configurations for subsequent applications. As a practical example, workers operating near power lines may be exposed to a relatively uniform field, as they are typically required to maintain a certain safe distance from the central field source (the power line). These details are added in the revised manuscript.
- In real-world conditions, CIEDs are in direct contact with conductive tissue within the body, forming a loop as described by the author, which allows voltage to be induced according to the theory presented in Section II. However, the measurement setup shown in Fig. 3 does not appear to replicate this environment adequately. The authors should provide detailed information on the setup, including the material of the phantom case, the type of fluid used inside, and the positioning and orientation of the device. Furthermore, the authors should discuss how these setup parameters might introduce discrepancies or errors compared to actual physiological conditions. The reviewer questions whether this setup can accurately measure the induced voltage as predicted by Equation (1).
A PVC container (L × W × H: 310 mm × 170 mm × 220 mm) filled with saline solution, with an electrical conductivity of 0.2 S/m, was used as a phantom to simulate human body tissue. We used an electrical conductivity of 0.2 S/m for the phantom saline solution as the average conductivity of the body in accordance with ICNIRP. In real experimental condition, the distribution of the phantom solution may vary during the measurements. We ensured that the solution was well mixed in the interval of measurements and maintain the bias within ±0.02. (Highlighted in the revised manuscript)
In real-case conditions, the configurations may vary in many aspects due to individual physiological structures and parameters of devices and their implantations. The full range of diversity is not feasible to be practically accommodated in this study. In this work, we studied factors including sensing mode, implantation method, exposure frequency, and exposure orientation. The variations in these factors may significantly affect the interference. Thus, we focus primarily on examining how these factors exert their influence.
- As shown in Fig. 4, there is almost no noticeable difference between the results at angles of 20° and 30°, and no clear trend is observed. This lack of variation warrants further discussion. Additionally, the authors should clarify the rationale behind choosing only 20° and 30° as test angles. If the goal was to comprehensively assess angular dependence, a wider range of angles (e.g., 20°, 30°, 60°, 90°) should have been included.
The configurations with a certain exposure orientation angle are subjected to a reduction of effective induction area. With frontal exposure represents the worst-case scenario in orientation, angles of 20° and 30° were selected to examine whether the interference is affected by reduction of the loop's projection area. The results align with our assumption and the effect is more substantial in bipolar sensing mode. This likely occurs because the projected area of the angled configuration becomes too small to prevent significantly influence from perturbations in the conductive medium. The discussion is included in the revised manuscript.
- As noted in question (2), it remains questionable whether the measured and simulated values in this study follow the same trend as the calculated values. According to Table 1, there is a substantial discrepancy between the calculated results and the simulated/measured data. Given the simplicity of the tested pathway, such significant deviations further reinforce the reviewer’s concerns regarding the validity of the theoretical assumptions and the accuracy of the measurement setup.
The conservative approach provides general estimation of the induced voltage to guarantee protection to all possible configurations, however, some conditions in real-case scenarios may be neglected. Firstly, the reduction of effective induction area caused by the IPG is not taken into consideration in the calculation. Secondly, in real-case exposures, the current path in the conductive medium is distributed over a volume rather than being confined to a one-dimensional line. The spread of the current return path within the volume can be easily deformed or displaced by external influences, potentially reducing the effective induction area. The actual induction area is smaller than that of the theoretical approach. In the bipolar sensing mode, where the two current paths are located in close proximity, this effect may be more pronounced. (Highlighted in the revised manuscript) This may also reduce the induced voltages in the measurement compared to the theoretical calculation.
In this work, we aim to provide reliable verification of the impact of the various factors and insight into the levels of induced voltage. In addition, the induced voltages remained at micro-volt level. The dispersion is more noticeable in percentage but insignificant in absolute value.

Round 2
Reviewer 2 Report
Comments and Suggestions for Authors
No more comments